# An exploratory review of resiliency assessments after brain injury

Janna Griffioen[1,2]*, Nicole Gingrich[3], Courtney L. Pollock[1,4], Julia Schmidt[1,5]

**1** Rehabilitation Research Program, Centre for Aging SMART at Vancouver Coastal Health, Vancouver, British Columbia, Canada, **2** Graduate Program in Rehabilitation Sciences, University of British Columbia, Vancouver, Canada, **3** GF Strong Rehabilitation Centre, Vancouver Coastal Health, Vancouver, British Columbia, Canada, **4** Department of Physical Therapy, University of British Columbia, Vancouver, Canada, **5** Department of Occupational Science and Occupational Therapy, University of British Columbia, Vancouver, Canada

* janna.griffioen@gmail.com

## Abstract

### Background

People with brain injury can have lower resiliency compared to the general public. Yet, resiliency facilitates positive processes to negotiate adversity after brain injury. Therefore, measuring resiliency after a brain injury is important.

### Objective

The review aimed to (1) identify self-report resiliency outcome measures for use with people after acquired brain injury, using the process-based Traumatic Brain Injury Resiliency Model as the guiding conceptual framework, and (2) summarize the psychometric properties of the identified outcome measures and the utility of these measures in clinical rehabilitation.

### Method

The COSMIN guidelines for systematic reviews were followed to ensure appropriate reporting for each measure. Databases CINAHL, EMBASE, Medline, and PsychINFO were searched and independently reviewed by two people. Articles providing data on psychometric properties for measures of resilience for people with brain injury were retrieved. Psychometric properties and clinical utility (number of items, scoring details) were summarized.

### Results

Thirty-two articles were retrieved, including nine measures of resiliency: Acceptance and Action Questionnaire–Acquired Brain Injury, Confidence after Stroke Measure, Connor-Davidson Resilience Scale, Daily Living Self-Efficacy Scale, General Self-Efficacy Scale, Participation Strategies Self-Efficacy Scale, Resilience Scale, Robson Self-Esteem Scale, and the Stroke Self-Efficacy Questionnaire. All measures have acceptable to excellent psychometric properties in accordance with the COSMIN guidelines.

**Data Availability Statement:** All relevant data are within the paper and its Supporting Information files.

**Funding:** The authors disclosed receipt of the following financial support for the research,

authorship, and/or publication of this article: This work was supported by the Michael Smith Foundation for Health Research – Scholar award (Pollock) and the Social Sciences and Humanities Council - Project grant (Schmidt). The funders had no role in study design, data collection and analysis, decision to publish, or preparation of the manuscript.

**Competing interests:** The authors declared no potential conflicts of interest with respect to the research, authorship, and/or publication of this article.

## Conclusion

There are established measures of resiliency in brain injury rehabilitation. Future work may explore use of these measures in a clinical context and implementation of rehabilitation goals for improving resiliency after brain injury.

## Introduction

Individuals with brain injury (e.g., stroke and traumatic brain injury) often experience long-term changes in cognitive, physical, and psychosocial functioning [1]. Changes in functioning include reduced participation in meaningful everyday activities, decreased engagement in social roles [2–5], and decreased quality of life and mental health [6, 7]. For example, the presence of depression during rehabilitation has been associated with decreased health-related quality of life [8]. Long-term, consequences of living with brain injury can result in unemployment and decreased quality of life [9, 10]. While a brain injury can initiate adversity in many facets of life, resiliency can facilitate positive processes to meet adversity after brain injury [11]. As such, resiliency may be an important construct to measure in rehabilitation.

Resiliency is a process to negotiate cognitive, physical, and psychosocial challenges, and facilitate positive perspectives [12]. Resiliency can include the ability to appraise, adapt, and meet broad challenges associated with a brain injury [11]. The Traumatic Brain Injury Resiliency Model (TBIRM) describes processes (e.g., response to adversity, self-regulatory processes, environment) that can interact to achieve resiliency outcomes after brain injury [11]. Resiliency, as defined in the TBIRM, is "the combined interplay among a set of affective, behavioral, and cognitive protective factors and self-regulatory processes that enable individuals to negotiate or bounce back from adversity" [11]. According to the TBIRM, resilience encompasses an individual's trait or state, while resiliency relates to an individual's process and the many factors it influences [11].

Resilience, being thought of as a state, may be lower in brain injury populations compared to the general population [13]. Notably, lower resilience in traumatic brain injury (TBI) populations has been related to lower education and pre-injury substance use; as well as psychological distress and low quality of life experienced after a brain injury [14]. Rehabilitation focused on improving resilience has shown to improve health outcomes such as depression, anxiety, sleep, and post-traumatic stress disorder [15–17]. The process of resiliency in brain injury, introduced by the TBIRM, is a new concept with promising utility.

Clinical outcomes of brain injury rehabilitation commonly focus on impairments of the injury itself (e.g., cognitive and physical challenges). While clinical outcomes may indicate reduction of deficits, they do not consider the influence and importance of resiliency on health and social outcomes. Identifying resiliency measures aligning with the TBIRM conceptual framework is the first step to applying such measures in clinical practice and research. Once applied, the impact of interventions on resiliency can be understood.

The purpose of this review was to identify self-report outcome measures of resiliency, validated for use with people with brain injury. The review also aimed to identify the psychometric properties of the identified outcome measures and examine the utility of these measures in clinical rehabilitation.

## Methods

An exploratory review was conducted using the TBIRM as a conceptual framework with a specific focus on "resiliency-related outcomes" [11]. This includes aspects relating to one's

individual experience within their environmental context. Specifically, the TBIRM highlights three key resiliency-related outcomes including: Reconstructing Identity (e.g., accepting disability and creating new goals), Re-engaging in Activities (e.g., participation in normal activities), and Adjusting Participation Patterns and Preferences (e.g., accept and adapt) [11].

## Sources and search strategy

The COSMIN guidelines were followed for a systematic review literature search to ensure the literature search was thorough and valid [18]. Four electronic databases were comprehensively searched since inception: CINAHL (1982), EMBASE (1974), MEDLINE (1946), and PsychINFO (1967). Databases were chosen as they relate to intervention by allied health groups that report on brain injury outcome measures and capture a broad interdisciplinary perspective. Database searches took place in August 2022. Search terms were developed as indicated by COSMIN guidelines within the categories of (a) construct, (b) population, (c) clinical measure, and (d) psychometric properties of interest [S1 Table; 18]. S1 Table describes the search terms used for each of the four databases. Boolean terms ("AND", "OR", "NOT") were used to combine terms, and asterisks were used to include variants in spelling (e.g., "resilienc*"). Limiters for English, full-text, peer-reviewed articles were applied. A research librarian was consulted, and search terms adjusted to capture all outcome measures as they relate to resiliency in acquired brain injury rehabilitation. Due to the nature of the present review, the definitions of resilience and resiliency are delineated but included in the search to identify all outcome measures.

To obtain studies of resiliency, alternate terms were required in the database searches in order to broaden and increase the sensitivity of the search as the search term "resiliency" was not identified in most databases until the previous few years. For example, EMBASE created the term resilience in 2017 from the previous term "coping behaviour". Our search extracted outcome measures from databases since inception, and as such, alternate terms were used to maintain a rigorous search throughout the database history. The search terms self-efficacy, acceptance, and self-confidence were identified through terms used in the TBIRM, previous search terms used in the databases, librarian consultation and clinical judgement. Table 1 outlines the keyword definitions of the search terms. The search included terms for resiliency (e.g., resiliency, resilience, self-efficacy, acceptance, and self-confidence), population (acquired brain injury), clinical measure (self-reported outcome), and psychometric property (validity and utility).

**Table 1. Definitions of key terms.**

| Key Term | Definition |
|---|---|
| Resiliency | "The combined interplay among a set of affective, behavioral, and cognitive protective factors and self-regulatory processes that enable individuals to negotiate or bounce back from adversity" [6, p. 2709] |
| Resilience | "A trait or personal quality which allows individuals to rise above or do well in spite of adversity" [6, p. 2709] |
| Self-confidence | An affective component of resiliency; one's sense of assuredness in one's own self and abilities [6–8] |
| Self-efficacy | A behavioral component of resiliency; one's belief in their capacity to manage a situation and achieve their desired outcome [6, 9] |
| Acceptance | A cognitive component of resiliency; patients' acknowledgement that a problem is not likely to disappear, and it is better to adjust goals to accommodate available resources and constraints [6, 10] |

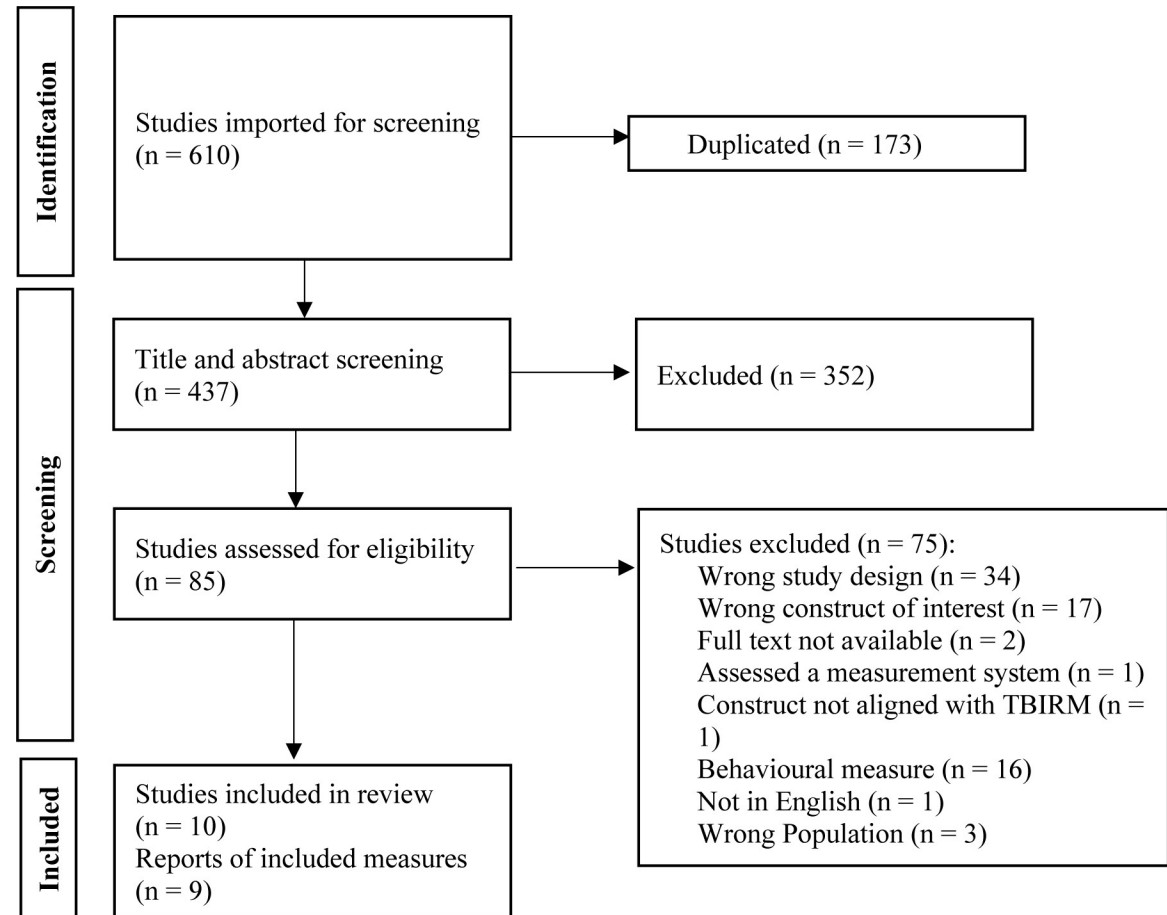

**Fig 1. Study selection PRISMA.** The PRISMA flow diagram for the systematic review detailing the number of abstracts obtained, screened, and reviewed on full texts. Manuscript exclusions are detailed in each stage of review.

A secondary search was run with the title of each clinical measure, and the psychometric terms (validity and utility) [Fig 1]. The secondary search aimed to retrieve additional psychometric information specific to each clinical measure [Fig 1].

## Inclusion and exclusion criteria

Article inclusion criteria were as follows: (a) the clinical measure assessed resiliency according to the TBIRM in adults with acquired brain injuries (e.g., TBI, stroke [19]); (b) the article included evaluation of at least one psychometric property of the given measure, even if the measure was not the focus; (c) the measure was a subjective self-report measure; (d) articles were in English, peer-reviewed full-texts.

Article exclusion criteria were as follows: (a) the measure was only validated for use with children or adolescents under the age of 18 given the different experiences of disability in a younger population [20, 21]; (b) the measure was not subjective in nature, and exclusively utilized objective, proxy, or informant report information; (c) the measure does not assess resiliency in accordance to the resiliency-related outcomes in the TBIRM. If the article included collections of clinical measures, then the article was excluded. Collections of measures were further screened for any clinical measures that were missed in the search.

### Data extraction and analysis

The search results from each of the four databases were merged into Covidence, a systematic review management platform. Title and abstract screening was completed by two independent reviewers (authors JG and JS). Conflicts were resolved with conversations. Full text review took place by two independent reviewers (JG and JS) with conflicts resolved through discussion.

Data on each measure was extracted by two independent reviewers including population validated for use, time post-injury (i.e., community dwelling or during rehabilitation), clinical utility (i.e., number of items, size of Likert Scale, availability of clinical measure), and content of questions. Measures were evaluated for construct validity and reliability by the articles identified, making the risk of bias tied to the psychometric properties reported on the measure itself. The methods followed the PRISMA guidelines for systematic reviews and abstracts [S2–S4 Tables].

To understand how each measure aligned with the TBIRM, two authors (JG and JS) independently reviewed the purpose of the measure and individual items of each measure. Authors then independently identified which of the three resiliency-related outcomes of the TBIRM the measure was most aligned (e.g., Reconstructing Identity, Re-engaging in Activities, and Adjusting Participation Patterns and Preferences). Where the purpose and items within each measure also were secondarily aligned to another resiliency-related outcomes, these measures were categorized as having a secondary alignment to that resiliency-related outcome.

## Results

The search yielded 417 articles, leaving 305 after 112 duplicates were removed (see Fig 1). After title and abstract screening, 65 articles remained for full-text review. There were 33 articles excluded based on the inclusion criterion, with a remaining 32 articles included in this review. These articles collectively retrieved 9 clinical measures of resiliency (see Fig 1, PRISMA diagram, for reasons of exclusion). One clinical measure, the Connor-Davidson Resilience Scale, was found from screening the RAND and Review of Positive Psychology Outcome Measures, but not found in the database search [1, 22, 23].

The measures differed in the specific clinical population which the measure has been validated for use, the format of the measure, and how the measure relates to resiliency (see Table 2). All measures used a similar format of providing statements for participants to rate their level of agreement or disagreement. The measures range from 10–35 statements.

The psychometric properties of each clinical measure vary. Clinical measures in this review report psychometric properties to guide clinicians and researchers to choose appropriate measures based on the brain injury sub-population. Table 2 provides a summary description of the psychometric properties of each clinical outcome measure; this does not constitute an independent evaluation of psychometric properties.

### TBIRM resiliency-related outcomes

Measures differed in relation to the resiliency-related outcomes of the TBIRM (see Table 3). After analyzing the purpose and individual items of each measure, we identified three measures that aligned with the resiliency-related outcome of Reconstructing Identity, three measures that aligned with the outcome of Re-engaging in Activities, and two measures that aligned with Adjusting Participation Patterns or Preferences. Below is a description of how each measure identified assesses resiliency-related outcomes from the TBIRM.

**Reconstructing identity.** As per the TBIRM, reconstructing identity is the ability to recognize the large change TBI enacts upon identity [11]. The act of reconstructing identity

**Table 2. Self-report resiliency measures summary and psychometric properties.**

| Measure | Content and example of Questions | Clinical Utility | Psychometric Properties | |
| --- | --- | --- | --- | --- |
| | | | **Participants tested** | **Psychometric property** |
| **Acceptance and Action Questionnaire–Acquired Brain Injury** | Assessing beliefs explicitly in the context of brain injury. Item: "My brain injury defines me." | 15 items; 3 Subscales: Reactive Avoidance, Denial, and Active Acceptance; 5-point Likert Scale: (0) "not at all true" to (4) "very true"; higher scores reflect greater psychological inflexibility; Items are listed in article, must submit request for copy of full assessment [13]. | Kortte et al., 2009 **n = 139** Spinal cord dysfunction secondary to spinal cord injury, Guillain-Barre, or multiple sclerosis (spinal cord): n = 82; Ischemic or hemorrhagic strokes (stroke): n = 23; Amputations: n = 16; Hip or knee replacements (orthopedic): n = 18 Age (years): Mean = 54.90, SD = 18.72 Male: 60.4%; Female: 39.6% | Internal consistency a = 0.70 (acceptable) |
| | | | | Construct validity Facilitation of Action Factor: factor loadings = 0.16 to 0.80 Evaluation of Affect Factor: factor loadings = -0.44 to 0.69 |
| | | | | Predictive validity Life satisfaction (during participation) ß = -0.45, p<0.000 Life satisfaction (3mo f/u) ß = -0.40, p<0.001 Level of handicap ß = -0.20, p<0.014 Social integration ß = -0.23, p<0.012 Rehabilitation engagement (during participation) ß = -0.13, ns Rehabilitation engagement (3mo f/u) ß = -0.07, ns |
| | | | Whiting, Diane L; Deane, Frank P; Ciarrochi, Joseph; McLeod, Hamish J; Simpson, Grahame K, 2015 **n = 150** Severe traumatic injury: n = 117; Brain tumour: n = 11; Hypoxic injury: n = 9; Cerebrovascular accident: n = 13 Age (years): Mean = 38.1, SD = 13.7 Male: 77.3% Female: 22.7% | Internal consistency Factor 1 (Reactive Avoidance): a = 0.89 (good) Factor 2 (Denial): a = 0.38 (unacceptable) Factor 3 (Active Acceptance): a = 0.46 (unacceptable) |
| | | | | Test-retest reliability Factor 1 (Reactive Avoidance): ICC = 0.92 (95% CI 0.86 to 0.95) (high degree of reliability) Factor 2 (Denial): ICC = 0.75 (95% CI 0.60 to 0.85) (not reliable) Factor 3 (Active Acceptance): ICC = 0.68 (95% CI 0.49 to 0.80) (not reliable) |
| | | | | Construct validity Scores on Factor 1: good Scores on Factors 2 and 3: weak |
| **Confidence after Stroke Measure** | Assessing general beliefs, yet informed by disability experience. Mix of positively and negatively positioned questions. Self-confidence subscale: "I feel robbed of my identity" Positive attitude subscale: "I believe I have inner strength" Social confidence subscale: "I feel scared to go out" | 27 items; 3 factors: Self-Confidence, Positive Attitude and Social Confidence; 4-point Likert scale: (0) low confidence to (3) high confidence; Assessment available as supplemental material to article [14]. | Horne, Jane C.; Lincoln, Nadina B.; Logan, Pip A., 2017 **n = 202** Healthy elderly population: n = 101; Stroke respondents: n = 101 Age (years): Mean = 70.1, SD = 13.3 Male: 45.5% Female: 54.5% | Internal consistency (27-item questionnaire) Participant Groups: Stroke participants: a = 0.92 (excellent) Healthy elderly participants: a = 0.90 (excellent) Sub-scales: Self-confidence α = 0.89 (good) Positive attitude α = 0.82 (good) Social confidence α = 0.88 (good) |
| | | | | Test-retest reliability Spearman's correlation r = 0.85, p = 0.001 (good temporal stability) Wilcoxon Signed Rank Test p = 0.04 |
| | | | | Face validity Good |
| | | | | Content validity Good |
| | | | | Convergent validity Spearman's correlation between 27-item Confidence after Stroke Measure and Stroke Self-Efficacy Questionnaire: r = 0.77, p = 0.001 |

(*Continued*)

**Table 2.** (Continued)

| Measure | Content and example of Questions | Clinical Utility | Psychometric Properties | |
|---|---|---|---|---|
| | | | **Participants tested** | **Psychometric property** |
| **Connor-Davidson Resilience Scale** | Assessing general beliefs, not specific to health or disability. Positively positioned questions. Item: "I am able to adapt when changes occur" Item: "I tend to bounce back after illness, injury, or hardship" | 25 items, 10 item, and 2 item versions; 5-point Guttmann-type scale: (0) "not true at all" to (4) "true nearly all of the time"; Total scores range from 0–100, higher scores indicate higher resilience; Must contact copyright holders for copy of clinical measure and instructions for administration. Sample manual available on measure's website. | Stoner et al., 2015 **n = 806** Random-digit dial general population (non-help-seeking), primary care recipients, psychiatric outpatients, GAD and PTSD Age (years): Mean = 43.8, SD not reported Sex not reported | Internal consistency $\alpha$ = 0.89 (good) |
| | | | | Test-retest reliability ICC = 0.87 (good) |
| | | | | Convergent validity With SSS: r = 0.36, p<0.0001 (high significant positive correlation) With PSS-10: r = -0.76, p<0.001 (significant negative correlation) With SVS: r = -0.32, p<0.0001 (high significant negative correlation) With SDS: r = -0.62, p<0.0001 (high significant negative correlation) |
| | | | | Criterion validity With Kobasa hardiness: r = 0.83, p<0.0001 (significant positive correlation) |
| | | | | Sensitivity to change Effect of time (F = 17.36; d.f. 1, 46; p<0.0001) Interaction between time and response category (F = 12.87; d.f. 2, 47; p<0.001) Both indicate scores increased with overall clinical improvement |
| **Daily Living Self-Efficacy Scale** | Assessing general beliefs, yet informed by disability experience on aspects of both psychosocial functioning and ADLs. Mix of positively and negatively positioned statements. Psychosocial functioning subscale: "Do something that helps me feel better when feel down" ADL subscale: "Look after my finances" | 12 items; 2 subscales: (1) Self-efficacy for psychosocial functioning and (2) self-efficacy for activities of daily living (ADL); 10-point Likert scale with 10-unit intervals from (0) "cannot do at all" to (100) "highly certain can do" referring to statement "I am confident that I can. . ."; Total scores range from 0–100, higher scores indicating higher self-efficacy; Items are listed in article, must submit request for copy of full clinical measure [15]. | Maujean, Annick; Davis, Penelope; Kendall, Elizabeth; Casey, Leanne; Loxton, Natalie, 2014 **n = 424** Stroke survivors: n = 259; Control group (without stroke or any brain injury): n = 165 Age (years): Mean = 65.3, SD = 12.7 Male: 46.5% Female: 53.5% | Internal consistency Full sample: Total scale a = 0.95 (excellent) Psychosocial functioning a = 0.94 (excellent) Activities of daily living a = 0.91(excellent) Stroke Group: Total scale a = 0.95 (excellent) Psychosocial functioning a = 0.93 (excellent) Activities of daily living a = 0.91 (excellent) Non-stroke Group: Total scale a = 0.88 (good) Psychosocial functioning a = 0.90 (excellent) Activities of daily living a = 0.64 (questionable) |
| | | | | Test-retest reliability ICC_agreement of all items = 0.78–0.98 (good-excellent temporal stability) |
| | | | | Convergent validity DLSES and Patient Competency Rating Scale—participants' ratings: n = 0.74, p<0.001 (high positive correlation) DLSES and Generalized Self-Efficacy Scale: r = 0.56, p<0.001 (moderate positive correlation) DLSES and Patient Competency Rating Scale—carers' ratings: r = 0.59, p<0.001 (moderate positive correlation) |
| | | | | Discriminant validity DLSES and TICS-M: r = 0.11 (non-significant correlation) DLSES and Barthel Index: r = 0.28 (very low significant positive correlation) |

*(Continued)*

**Table 2.** (Continued)

| Measure | Content and example of Questions | Clinical Utility | Psychometric Properties | |
|---|---|---|---|---|
| | | | **Participants tested** | **Psychometric property** |
| **General Self-Efficacy Scale** | Assessing general beliefs, not specific to health or disability. All questions are positively positioned. Item: "I can usually handle whatever comes my way" | 10 statements; Scale from (1) "not at all true" to (4) "exactly true"; Total scores range from 10–40, higher scores indicate greater sense of general self-efficacy; Scale is publicly available via the clinical measure website | Carlstedt, Emma, Eva Månsson Lexell, Hélène Pessah-Rasmussen, and Susanne Iwarsson. 2015 **n = 34** Infarction (stroke): n = 33; Hemorrhage (stroke): n = 1 Age (years): Mean = 68.1, SD not reported Male: 61.8% Female: 38.2% | Internal consistency a = 0.92, 95% CI 0.86 to 0.95 (excellent) Test-retest reliability $ICC_{2,1}$ = 0.82 (95% CI 0.67 to 0.90) Systematic/random differences d = -0.68 (95% CI -2.23 to 0.88) |
| **Participation Strategies Self-Efficacy Scale** | Assessing general beliefs, yet informed by disability experience. Item: "Adapt home activities to do what you want/need to" Item: "Strategize falling or fear of falling in the community" Item: "Access services to help stay in home" | 35 items; 10-point Likert scale from (1) "not at all confident" to (10) "totally confident"; Rating confidence/ self-efficacy in 6 participation domains in home, work and community referring to statement "I am confident that I can…";; Items are listed in article, must email to request copy of full clinical measure [16]. | Lee, Danbi; Fogg, Louis; Baum, Carolyn M.; Wolf, Timothy J.; Hammel, Joy, 2018 **n = 166** Mild to moderate stroke (NIH stroke scale <16) Age (years): Mean = 56.5, SD = 10.33 Male: 50.6% Female: 49.4% | Internal consistency Home management: a = 0.904 (excellent) Organizing at home: a = 0.861 (good) Community management: a = 0.926 (excellent) Work management: a = 0.926 (excellent) Community service management: a = 0.907 (excellent) Communication management: a = 0.884 (good) *High Cronbach's alpha values may suggest that some items are redundant |
| **Resilience Scale** | Assessing general beliefs, not specific to health or disability. Includes positively positioned statements. Item: "I usually manage one way or another" | 25 items; 14 item short version also available (RS14); 7-point Likert Scale from (1) "disagree" to (7) 'agree'; Higher scores reflecting greater resilience; Must purchase clinical measure via the clinical measure website. | Losoi, Heidi, Noah D. Silverberg, Minna Wäljas, Senni Turunen, Eija Rosti-Otajärvi, Mika Helminen, Teemu Miikka Artturi Luoto, Juhani Julkunen, Juha Öhman, and Grant L. Iverson. 2015 **n = 113** Group 1: mild traumatic brain injury group n = 74 CT-imaged head injury patients Age (years): Mean = 37.0, SD = 11.8 Male: 61%; Female: 39% Group 2: Trauma control group n = 39 Age (years): Mean = 39.7, SD = 12.1 Male: 49%; Female: 51% | Internal consistency Resilience Scale: a = 0.91 to 0.93 (excellent) for mTBI group a = 0.88 to 0.95 (good-excellent) for controls Resilience Scale-14: a = 0.88 to 0.93 (good-excellent) for mTBI group a = 0.86 to 0.94 (good-excellent) for controls Test-retest reliability RS across studies: 0.67–0.84 RS and RS-14 across studies: 0.66 to 0.80 Content validity Strong Concurrent validity Strong |

(*Continued*)

**Table 2.** (Continued)

| Measure | Content and example of Questions | Clinical Utility | Psychometric Properties | |
|---|---|---|---|---|
| | | | **Participants tested** | **Psychometric property** |
| **Robson Self-Esteem Scale** | Assessing general beliefs, not specific to health or disability. Mix of positively and negatively positioned statements.<br>Item: "If I really try, I can overcome most of my problems"<br>Item: "I can never seem to achieve anything worthwhile" | 30 items; 8-point Likert scale from (0) "completely disagree" to (7) "completely agree"; 4 factors: Self-Worth, Self-Regard, Self-Efficacy, Confidence; Administration time 10 min.; Items are listed in article, must email for copy of full clinical measure [Longworth]; Also made available online under the name "Robson Self-Concept Questionnaire" [12]. | Longworth, Catherine; Deakins, Joseph; Rose, David; Gracey, Fergus, 2018<br>**n = 80**<br>TBI: n = 54; Stroke: n = 18; Encephalitis: n = 3; Hypoxia: n = 2; Meningitis: n = 1; Other: n = 2<br>Age (years): Mean = 35.55, SD = 10.83<br>Male: 67.5%<br>Female: 32.5% | Internal consistency<br>$\alpha = 0.89$ (good)<br>Guttmann split half reliability = 0.75 (good)<br>Factors:<br>Self-worth: a = 0.82 (good)<br>Self-regard: a = 0.86 (good)<br>Self-efficacy: a = 0.72 (acceptable)<br>Confidence and determinism: a = 0.6 (questionable) |
| | | | | Construct validity<br>Kaiser-Meyer-Olkin (KMO) measure of sampling adequacy = 0.79<br>Bartlett's Test of Sphericity: $p<0.001$<br>Haitovsky test: $p<0.001$<br>Factor correlation matrix: (all not significant)<br>Self-worth and Self-regard = -0.36<br>Self-worth and Self-efficacy = -0.34<br>Self-worth and Confidence = 0.28<br>Self-regard and Self-efficacy = 0.34<br>Self-regard and Confidence = -0.24<br>Self-efficacy and Confidence = -0.23 |
| | | | | Factorial validity<br>Self-Regard predicted HADS depression, accounting for 38% of variance, $R^2 = 0.38$, $F(4, 58) = 9.00$, $p<0.001$, $\beta = -0.38$, $p = 0.01$<br>Two factor model:<br>Self-Worth ($\beta = -0.39$, $p<0.01$) and Self-Efficacy ($\beta = -0.30$, $p<0.05$) significantly predicted HADS anxiety, accounting for 44% of the variance, $R2 = 0.44$, $F(4, 58) = 11.26$, $p<0.001$ |
| **Stroke Self-Efficacy Questionnaire** | | 13 items; 0–10 scale from (0) "not at all confident" to (10) "very confident" for a variety of tasks; <15 min.; Clinical measure available at the end of article [18]. | Partridge, Cecily; Reid, Fiona; Jones, Fiona, 2008<br>**n = 112**<br>Adults with stroke<br>Sex not reported<br><u>Stage I</u><br>**n = 15**<br>Age (years): Mean age and SD not reported<br><u>Stage II</u><br>**n = 40**<br>Age (years): Mean = 68.4, SD not reported<br><u>Stage III</u><br>**n = 57**<br>Age (years): Mean = 65.0, SD = 17.9 | Internal consistency<br>a = 0.90 (excellent) |
| | | | | Face validity<br>Ceiling effect for those with high degree of independence in activities of daily living and mobility, enabled 10 items to be removed from the list<br>Final 13-item Stroke Self-Efficacy Questionnaire had good face validity |
| | | | | Criterion validity<br>High compared with Falls Efficacy Scale, $r = 0.803$, $p<0.001$ |

No equipment or special training is required for any measure.

*CMIN/df = minimum discrepancy per degree of freedom

CT = computed tomography

CVI = core values index

ICC = inter-class correlation

HADS = Hospital Anxiety and Depression Scale

MNSQ = mean square

**Table 3. Checklist of constructs assessed by measures.**

| Measure | Reconstructing | Re-engaging | Adjusting |
|---|---|---|---|
| Acceptance and Action Questionnaire–acquired brain injury | ✓ | | ✓ |
| Confidence after Stroke Measure | | | ✓ |
| Connor-Davidson Resilience Scale | ✓ | | ✓ |
| Daily Living Self-Efficacy Scale | | ✓ | |
| General Self-Efficacy Scale | ✓ | | ✓ |
| Participation Strategies Self-Efficacy Scale | | ✓ | |
| Resilience Scale &Resilience Scale Brief | | | ✓ |
| Robson Self-Esteem Scale | ✓ | | |
| Stroke Self-Efficacy Questionnaire | ✓ | ✓ | |

happens by performing, accepting, and identifying goals [11]. Three measures assessed the construct of Reconstructing Identity and are described below.

The General Self-Efficacy Scale is a component of the Emotional Health section of the National Institute of Health Toolbox and was validated for traumatic brain injury and stroke populations at least one-year post-injury [24]. The Swedish version has also been validated for the stroke population with good to excellent test-retest reliability and internal consistency [25].

The Resilience Scale was developed for individuals 1, 6, and 12 months after mild TBI, and may be limited in its applicability to community-dwelling populations with other brain injury diagnoses such as stroke or TBI [26]. The Resilience Scale evaluates the construct of reconstructing identity by using positively positioned statements assessing priorities using language that probes identity.

Lastly, the Robson Self-Esteem Scale, [27] assesses the process of identity reconstruction through factors such as self-deprecation, self-respect, attractiveness, and self-respect/confidence [28]. This measure has been validated for use with adults aged 17–56 who have complex and continued difficulties at least 9 months post-brain injury [27]. The measure aligned with the construct of reconstructing identity by assessing general beliefs in predominantly negatively positioned questions.

Measures that had a secondary alignment to this outcome included the Confidence After Stroke Measure (described in full below).

**Re-engaging in activities.** The TBIRM defines re-engagement as resuming valued activities at a level equivalent to pre-injury functioning [11]. Outcome measures identified within the construct of Re-engaging Activities include the Acceptance and Action Questionnaire for Acquired Brain Injury, the Daily Self-efficacy Scale, Participation Strategies Self-Efficacy Scale, and the Stroke Self-Efficacy Scale, described below.

The Acceptance and Action Questionnaire–Acquired Brain Injury measure included 3 subscales, one of which is "Active Acceptance" [29]. In line with the definition from the TBIRM, acceptance in this measure is seen as an active process rather than a passive resignation. The measure identifies active acceptance through positive mood and relationship.

The Daily Living Self-Efficacy Scale assesses self-efficacy to return to daily life with two subscales: self-efficacy for psychosocial functioning and for activities of daily living [30]. The scale has been validated with individuals several years after stroke, and its authors suggest its utility for assessing readiness for community living [30]. The scale has been validated through good to excellent test-retest reliability, convergent validity, and discriminant validity [30]. The scale includes questions to identify the process of Re-engaging in Activity as an outcome of resiliency.

Participation Strategies Self-Efficacy Scale (PSSES) assesses self-efficacy for community-level participation, with subscales for home, work, social and community management and participation [31]. The measure was developed for those with mild to moderate stroke and has limited utility in sub-acute or inpatient rehabilitation settings due to its evaluation of self-efficacy for community level participation [31]. Participation levels are measured using a subjective assessment of ability to re-engage in activity.

The Stroke Self-Efficacy Questionnaire, looks at self-efficacy using "self-management" and "activities" subscales and was validated with individuals 4–24 weeks post-stroke, from acute stroke units and living in the community [32, 33]. The recommended utility is to assess confidence during stroke recovery and influence the approach taken by clinicians during rehabilitation [33]. The measure uses questions assessing the Re-engagement of Activity that subjectively assess capability of participating in activities of daily living.

The Confidence After Stroke Measure also had a secondary alignment with the Re-engagement of Activity outcome.

**Adjusting participation patterns and preferences.** The TBIRM describes Adjusting Participation Patterns and Preferences as both objective adjustment and subjective adjustment [11]. Objective adjustment could be participation in new goals, while subjective adjustment could be changing perception of participation after brain injury. Outcome measures identified in this construct include the Confidence After Stroke Measure, and the Connor Davidson Resilience Scale.

The Confidence After Stroke Measure was created to guide treatment, support rehabilitation after stroke, and determine if lack of confidence is a potential barrier to recovery [34]. The measure assesses participation patterns through subjective questions.

The Connor Davidson Resilience Scale also fit in with Adjusting Participation Patterns and Preferences. The Connor-Davidson Resilience Scale was validated for traumatic brain injury and stroke [35] as part of the National Institute of Health Toolbox [36]. The scale has been used widely with the traumatic brain injury population [12, 37–39]. The Connor-Davidson Resilience Scale is unspecific in terms of chronicity post-brain injury. The measure assesses both subjective and objective assessments of participation using positively positioned questions. The measure uses subjective phrases.

Three measures had a secondary alignment with this outcome: Acceptance and Action Questionnaire for Acquired Brain Injury, General Self-efficacy Scale, and Resilience Scale (all discussed above).

## Discussion

The present review included nine clinical measures aligning with the TBIRM outcome measures of resiliency constructs: Reconstructing Identity, Re-engaging in Activities, and Adjusting Participation Patterns and Preferences. Of the nine clinical measures of resiliency available for use with brain injury populations, no singular clinical measure captures all aspects of the TBIRM resiliency-related outcomes. This may reflect the complexity of resiliency and the need for further exploration of resiliency as an outcome following brain injury.

This review highlights that resilience is a relatively new term in library databases. As such, our search required use of terms relating to resilience. For example, CINAHL refers to 'hardiness' as an alternate term to resiliency, a term that is not aligned with the TBIRM. EMBASE recently created the subject heading of resilience in 2017 after previously using the term "coping behaviour". Notably, outcome measures identified in the present review were not validated until 2009. Resiliency frameworks, such as the TBIRM, may assist with future clinical intervention and development of assessments methods. Shifting from a state or trait approach (i.e.,

resilience) to a process perspective (i.e., resiliency) may greatly impact the field's ability to assess resiliency from a holistic viewpoint.

There was variability in the validation of measures to adult brain injury sub-populations (e.g., stroke, TBI) and time since injury. For example, some measures were only validated for stroke populations (i.e., Confidence After Stroke Measure), others were validated for different severities and chronicity across brain injury populations (i.e., Robson Self-Esteem Scale) [27, 34]. Future research could consider further developing psychometrics of measures for all brain injury population groups.

When considering clinical relevance, some measures included in our review may have more clinical utility. Similar to selecting measures for other constructs, clinicians should consider person-level aspects of the measure (e.g., population for which the measure has been validated and the stage of the person's recovery) as well as the accessibility of measures in clinical practice (e.g., cost of the measure, training required). For resiliency measures in particular, clinicians may want to consider how the measure aligns with specific constructs within the TBIRM, as presented in our review. In this way, future research can identify potential outcome measures for use alongside exploration of the processes of developing resiliency after brain injury.

Many clinical measures examine task-level domains (e.g., physical, cognitive, and emotional outcomes) important for discharge to the community and community reintegration [24, 40]. Such measures may not include important patient-oriented constructs, and how resiliency influences and shapes quality of life over time [41]. The inclusion of patient-oriented constructs are important, as qualitative findings suggest that people with brain injury have alternate outcome priorities for rehabilitation after brain injury, including regaining their sense of self, improving self-efficacy, and regaining confidence in their ability to enjoy a meaningful life [27, 32, 33, 42]. This gap in assessing patient-orientated outcomes may represent an important omission to address more explicitly during rehabilitation and community integration. Further development of patient-oriented resiliency measures could translate into clinical practice and may support clinicians and researchers to better understand the needs of people with brain injury in rehabilitation and community care [43].

## Limitations

This review had two main limitations. Due to the small number of measures identified and associated articles, measures could not be divided into type of injury (e.g., TBI versus stroke) and time of use post injury (e.g., acute versus chronic stage of injury). However, this review provides an overview of all measures of resiliency for a brain injury population, which may have broader generalizability. Only studies in English were included in our review which may have limited the scope of the findings, as the understanding of the concept of resiliency could vary across languages and cultures [44].

## Conclusion

This review identified and mapped the available measures of resiliency for the brain injury population on the conceptual framework of the TBIRM. Resiliency is an important construct to measure in acquired brain injury rehabilitation, with many promising measures. Further research is needed to explore the implementation of these measures in clinical practice and the potential development of patient goals and treatment plans to facilitate resiliency.

## Supporting information

**S1 Table. Search terms.**
(DOCX)

**S2 Table. PRISMA 2020 checklist for systematic reviews.**
(PDF)

**S3 Table. PRISMA 2020 checklist for abstracts.**
(PDF)

**S4 Table. Data extracted from primary research sources.**
(DOCX)

## Author Contributions

**Investigation:** Nicole Gingrich.

**Supervision:** Courtney L. Pollock, Julia Schmidt.

**Writing – original draft:** Janna Griffioen, Nicole Gingrich.

**Writing – review & editing:** Janna Griffioen.

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
