## [Decision Letter · Decision Letter 0]

9 May 2024

PONE-D-23-27474An exploratory review of resiliency assessments after brain injuryPLOS ONE

Dear Dr. Griffioen,

Thank you for submitting your manuscript to PLOS ONE. After careful consideration, we feel that it has merit but does not fully meet PLOS ONE’s publication criteria as it currently stands. Therefore, we invite you to submit a revised version of the manuscript that addresses the points raised during the review process.

We look forward to receiving your revised manuscript.

Kind regards,

Diphale Joyce Mothabeng, PhD

Academic Editor

PLOS ONE

Journal Requirements:

 [The authors disclosed receipt of the following financial support for the research, authorship, and/or publication of this article: This work was supported by the Michael Smith Foundation for Health Research – Scholar award (Pollock) and the Social Sciences and Humanities Council - Project grant (Schmidt). ].  

Additional Editor Comments:

Thank you for the manuscript submitted.

Kindly address the reviewers comments provided to improve the manuscript clarity and quality.

Reviewers' comments:

Reviewer's Responses to Questions

**Comments to the Author**

1. Is the manuscript technically sound, and do the data support the conclusions?

Reviewer #1: Yes

Reviewer #2: Yes

2. Has the statistical analysis been performed appropriately and rigorously? 

Reviewer #1: Yes

Reviewer #2: N/A

3. Have the authors made all data underlying the findings in their manuscript fully available?

Reviewer #1: Yes

Reviewer #2: Yes

4. Is the manuscript presented in an intelligible fashion and written in standard English?

Reviewer #1: Yes

Reviewer #2: Yes

5. Review Comments to the Author

Reviewer #1: An exploratory review of resiliency assessments after brain injury

Thank you for the opportunity to review the manuscript. I reviewed this manuscript carefully with a great interest. I respectfully provided my comments below

The abstract does not clearly lay out the objective/question of the systematic review and an overview of the methods that will be used.

Attach the strategy search for each database

Explain data management and data synthesis.

How is the risk of bias in studies checked?

Reviewer #2: The systematic review aimed to identify measures of resilience for individuals following acquired brain injury, employing the process-based Traumatic Brain Injury Resiliency model. The review yielded nine studies, each examining various measurements of resilience. These measurements were assessed for their psychometric properties and clinical utility.

In the abstract, it is suggested to provide a more detailed explanation of the nine resilience measurements identified, along with a summary of their respective psychometric properties.

In the conclusion, the authors discuss three resilience outcomes delineated by the Traumatic Brain Injury Resiliency Model. It is recommended to incorporate these three outcomes into the methodological guidelines mentioned in the study.

The introduction and method sections were found to be well-written and comprehensive.

To enhance clarity, the results section could be divided into two subsections, each accompanied by relevant tables. The first subsection could outline the general characteristics of the included studies, such as country of origin, sampling settings, participant demographics, and details of the questionnaires utilized. The second subsection could focus on presenting the identified outcomes, including the questionnaires employed, their dimensions and sub-dimensions, the number of items within each, and the overall number of items per questionnaire. Psychometric properties should be presented alongside these findings.

Overall, these revisions aim to improve the organization and presentation of the study's results, ensuring a more informative and accessible review for readers.

6. PLOS authors have the option to publish the peer review history of their article (what does this mean?). If published, this will include your full peer review and any attached files.

Reviewer #1: **Yes: **Razieh Bandari

Reviewer #2: **Yes: **Leila jahangiry

---

## [Author Response · Author response to Decision Letter 0]

8 Jul 2024

Editor Feedback Author Response

Please state what role the funders took in the study. If the funders had no role, please state: “The funders had no role in study design, data collection and analysis, decision to publish, or preparation of the manuscript.” If this statement is not correct you must amend it as needed. Please include this amended Role of Funder statement in your cover letter; we will change the online submission form on your behalf. 

 We have now included a statement in our cover letter to indicate that the funders had no role in the study design, data collection and analysis, decision to publish, or preparation of the manuscript.

Please include captions for your Supporting Information files at the end of your manuscript, and update any in-text citations to match accordingly. Please see our Supporting Information guidelines for more information: http://journals.plos.org/plosone/s/supporting-information. 

 We have now provided supporting information files at the end of the manuscript and updated in-text citations.

Responses to Reviewers

Responses to Reviewer #1:

The abstract does not clearly lay out the objective/question of the systematic review and an overview of the methods that will be used. We have changed the abstract according to reviewer feedback with clear background, objective, and overview of the methods. 

 Abstract:

 Background: People with brain injury can have lower resiliency compared to the general public. Yet, resiliency can facilitate positive processes to negotiate adversity after brain injury. It is therefore important to measure resiliency after a brain injury. Objective: The review aimed (1) to identify self-report resiliency outcome measures for use with people after acquired brain injury, using the process-based Traumatic Brain Injury Resiliency Model as the guiding conceptual framework, and (2) to summarize the psychometric properties of the identified outcome measures and the utility of these measures in clinical rehabilitation. Method: The COSMIN guidelines for systematic reviews were followed to ensure appropriate reporting for each measure. Databases CINAHL, EMBASE, Medline, and PsychINFO were searched and independently reviewed by two people.

Attach the strategy search for each database. 

 A comprehensive search strategy for each database has been added to Supplemental Material 1. 

Explain data management and data synthesis. 

 We have now included information on the data management and synthesis, in accordance with the PRISMA guidelines. 

 Page 8, paragraph 3: 

 “The search results from each of the four databases were merged into Covidence, a systematic review management platform. Title and abstract screening was completed by two independent reviewers (authors JG and JS). Conflicts were resolved with conversations. Full text review took place by two independent reviewers (JG and JS) with conflicts resolved through discussion.”

How is the risk of bias in studies checked?

 We thank the reviewer for raising this comment. Our review focused on the quality and content of resiliency assessments themselves, as opposed to the quality of the studies from which the assessments were identified. We therefore chose to provide psychometric properties of each assessment to provide validity and reliability information on each assessment.

Responses to Reviewer #2: 

In the abstract, it is suggested to provide a more detailed explanation of the nine resilience measurements identified, along with a summary of their respective psychometric properties. 

 We thank the reviewer for the suggestion to add more detail of the results from the manuscript to be added to the abstract. Further detail, including the resiliency measures identified and their psychometric properties, has been added into the results section of the abstract. 

 Abstract

 Background: People with brain injury can have lower resiliency compared to the general public. Yet, resiliency can facilitate positive processes to negotiate adversity after brain injury. It is therefore important to measure resiliency after a brain injury. Objective: The review aimed (1) to identify self-report resiliency outcome measures for use with people after acquired brain injury, using the process-based Traumatic Brain Injury Resiliency Model as the guiding conceptual framework, and (2) to summarize the psychometric properties of the identified outcome measures and the utility of these measures in clinical rehabilitation. Method: The COSMIN guidelines for systematic reviews were followed to ensure appropriate reporting for each measure. Databases CINAHL, EMBASE, Medline, and PsychINFO were searched and independently reviewed by two people.

In the conclusion, the authors discuss three resilience outcomes delineated by the Traumatic Brain Injury Resiliency Model. It is recommended to incorporate these three outcomes into the methodological guidelines mentioned in the study. 

 We have now included more explanation of the resiliency-related outcomes in the methods section. We have also aligned our results section to the three resiliency-related outcomes. 

 Page 5, paragraph 3:

 Specifically, the TBIRM highlights three key resiliency-related outcomes including: Re-engaging in Activities (e.g., participation in normal activities), Adjusting Participation Patterns and Preferences (e.g., accept and adapt), and Reconstructing Identity (e.g., accepting disability and creating new goals) [11]. 

 Page 9, paragraph 2:

 To understand how each measure aligned with the TBIRM, two authors (JG and JS) independently reviewed the purpose of the measure and individual items of each measure. Authors then independently identified which of the three resiliency-related outcomes of the TBIRM the measure was most aligned (e.g., Reconstructing Identity, Re-engaging in Activities, and Adjusting Participation Patterns and Preferences). 

To enhance clarity, the results section could be divided into two subsections, each accompanied by relevant tables.

The first subsection could outline the general characteristics of the included studies, such as country of origin, sampling settings, participant demographics, and details of the questionnaires utilized. The second subsection could focus on presenting the identified outcomes, including the questionnaires employed, their dimensions and sub-dimensions, the number of items within each, and the overall number of items per questionnaire. Psychometric properties should be presented alongside these findings.

 We thank the reviewer for the suggestion and agree that the presentation of results can be improved. We have now modified the tables to better showcase the results. Table 2 describes the psychometric properties of each assessment included in the review. Table 3 and corresponding second section focuses on the TBIRM and where each measure falls within the subsections of Reconstructing Identity, Re-engaging in Activities, and Adjusting Participation Patterns and Preferences. 

 Page 11 – 21, Table 2. Self-Report Resiliency Measures Summary and Psychometric Properties

 Page 22, Table 3. Checklist of Constructs Assessed by Measures

---

## [Decision Letter · Decision Letter 1]

19 Aug 2024

PONE-D-23-27474R1An exploratory review of resiliency assessments after brain injuryPLOS ONE

Dear Dr. Griffioen,

Thank you for submitting your manuscript to PLOS ONE. After careful consideration, we feel that it has merit but does not fully meet PLOS ONE’s publication criteria as it currently stands. Therefore, we invite you to submit a revised version of the manuscript that addresses the points raised during the review process. The reviewers are supportive of publication of your manuscript though one reviewer has requested some additional changes and clarifications which include some greater clarity of the definition of key terms, a clearer rationale for the study and reducing the detailed descriptions of the instruments in the discussion.

We look forward to receiving your revised manuscript.

Kind regards,

Belinda J Gabbe, PhD

Academic Editor

PLOS ONE

Journal Requirements:

Reviewers' comments:

Reviewer's Responses to Questions

**Comments to the Author**

1. If the authors have adequately addressed your comments raised in a previous round of review and you feel that this manuscript is now acceptable for publication, you may indicate that here to bypass the “Comments to the Author” section, enter your conflict of interest statement in the “Confidential to Editor” section, and submit your "Accept" recommendation.

Reviewer #1: All comments have been addressed

Reviewer #3: (No Response)

2. Is the manuscript technically sound, and do the data support the conclusions?

Reviewer #1: Yes

Reviewer #3: Yes

3. Has the statistical analysis been performed appropriately and rigorously? 

Reviewer #1: Yes

Reviewer #3: Yes

4. Have the authors made all data underlying the findings in their manuscript fully available?

Reviewer #1: Yes

Reviewer #3: (No Response)

5. Is the manuscript presented in an intelligible fashion and written in standard English?

Reviewer #1: Yes

Reviewer #3: Yes

6. Review Comments to the Author

Reviewer #1: The authors have answered all the questions correctly, thanks to the research team.

In my opinion, this article is acceptable

Reviewer #3: Introduction

• Paragraph 1 sentence 3: “greater impairment” is vague and would benefit from clarity.

• Sentence 4: specify the direction of quality of life changes.

• Given resiliency is the key construct, an explicit definition is important to include at some early point in the introduction. This is included at the start of the third paragraph but is warranted earlier on, prior to any discussion of this construct.

• Paragraph 2: Ensure you have defined the acronym “TBI”.

• Paragraph 2 final sentence: please add a reference.

• Paragraph 3 final sentence: “resiliency relates to the many factors contributing to the process of resiliency” is unclear. This definition is circular, in that resiliency is described as contributing to resiliency. Re-wording this for clarity if recommended.

• You would benefit from a clearer rationale leading up the aims of this study, to more directly highlight why measures need to be identified and reviewed. It is noted that brain injury rehabilitation does not commonly address resiliency. This could be due to several factors, such as higher importance placed on other domains, lack of time, and/or lack of standardised procedures and tools. Your introduction has well established the importance of resiliency and enhancing this – what does the role of a measure play? What are the current gaps in our knowledge and use of measures? Your study identified 9 measures that I assume have been used within research and perhaps clinical settings, therefore more explicit rationale around why this review is necessary is needed.

Sources and Search Strategy

• This is well detailed.

Inclusions and Exclusion criteria

• “Measures that were used on acquired brain injury population (e.g., TBI stroke) were included given similar experiences with disability after injury”. This is unclear, as based on the introduction my understanding was that the study was focussing on brain injury. Please clarify, perhaps within your aims, whether brain injury or broader disability was the focus.

Results

• “The measures are reasonable in length for ease of use clinically, in a range from 10-35 statements”. What is the basis for this claim? It is unclear whether this is evidence-based or the authors’ interpretation of “reasonable in length”.

• The results section reads well, however can be presented in a briefer format. At present, there is a lot of detail describing each scale and this in some cases duplicates information presented in Table 2, such as example items and the population in which the scale has been validated. Text and tables should complement rather than duplicate the other. I feel the lengthy description of each scale is not needed in the latter part of the results section; this could focus more on how each scale maps onto the construct.

Discussion

• Line 306: “While some important factors and experiences overlap among subpopulations within brain injury (e.g., stroke versus traumatic brain injury)”. This claim would benefit from specific examples of factors.

• As per my earlier comment on the introduction, your discussion would benefit from more explicit note around why a valid tool is needed that addresses all aspects of resiliency. What role does a measure play in clinical settings and why is this needed?

• Limitations: Make sure there is evidence to support your claims. It is important to not just list limitations, but also explain how. “…as resiliency could vary across languages and non-English speaking regions” – is there evidence to support that resiliency might have cultural/language differences?

7. PLOS authors have the option to publish the peer review history of their article (what does this mean?). If published, this will include your full peer review and any attached files.

Reviewer #1: No

Reviewer #3: No

---

## [Author Response · Author response to Decision Letter 1]

9 Oct 2024

We appreciate the opportunity to engage with the reviewer's comments and have edited the manuscript accordingly. Revisions were made regarding the positioning of the review within current and future literature, and increased clarity in sections identified. A point by point response to reviewer feedback is attached to our revised submission.

---

## [Editor Report · Decision Letter 2]

28 Oct 2024

PONE-D-23-27474R2An exploratory review of resiliency assessments after brain injuryPLOS ONE

Dear Dr. Griffioen,

Thank you for submitting your manuscript to PLOS ONE. After careful consideration, we feel that it has merit but does not fully meet PLOS ONE’s publication criteria as it currently stands. Therefore, we invite you to submit a revised version of the manuscript that addresses the points raised during the review process.

 **The changes made in the revised version of the manuscript have clarified and addressed many of the reviewer comments.  Some further changes to improve clarity and reduce duplication of material have been requested.**

We look forward to receiving your revised manuscript.

Kind regards,

Belinda J Gabbe, PhD

Academic Editor

PLOS ONE
---

## [Author Response · Author response to Decision Letter 2]

2 Dec 2024

The submitted edits now reflect the journal requirements of a complete and correct reference list.

---

## [Editor Report · Decision Letter 3]

4 Dec 2024

An exploratory review of resiliency assessments after brain injury

PONE-D-23-27474R3

Dear Dr. Griffioen,

We’re pleased to inform you that your manuscript has been judged scientifically suitable for publication and will be formally accepted for publication once it meets all outstanding technical requirements.

Kind regards,

Belinda J Gabbe, PhD

Academic Editor

PLOS ONE
---

## [Editor Report · Acceptance letter]

10 Dec 2024

PONE-D-23-27474R3 

PLOS ONE

Dear Dr. Griffioen, 

I'm pleased to inform you that your manuscript has been deemed suitable for publication in PLOS ONE. Congratulations! Your manuscript is now being handed over to our production team.

Kind regards, 

on behalf of

Professor Belinda J Gabbe 

Academic Editor

PLOS ONE